# Analysis of the Influence of Waste Seashell as Modified Materials on Asphalt Pavement Performance

**DOI:** 10.3390/ma15196788

**Published:** 2022-09-30

**Authors:** Guopeng Fan, Honglin Liu, Chaochao Liu, Yanhua Xue, Zihao Ju, Sha Ding, Yuling Zhang, Yuanbo Li

**Affiliations:** 1National Engineering Research Center of Highway Maintenance Technology, Changsha University of Science & Technology, Changsha 410114, China; 2Henan Transport Investment Group Co., Ltd., Zhengzhou 450016, China; 3Wuhan Hanyang Municipal Construction Group Co., Ltd., Wuhan 430050, China

**Keywords:** asphalt pavement, seashell powder, modified asphalt, high- and low-temperature performance, water stability

## Abstract

An increasing amount of waste seashells in China has caused serious environmental pollution and resource waste. This paper aims to solve these problems by using waste seashells as modified materials to prepare high-performance modified asphalt. In this study, seashell powder (SP) and stratum corneum-exfoliated seashell powder (SCESP) were adopted to prepare 10%, 20% and 30% of seashell powder-modified asphalt (SPMA) and stratum corneum-exfoliated seashell powder-modified asphalt (SCESPMA) by the high-speed shear apparatus, respectively. The appearance and composition of two kinds of SPs were observed and determined by the scanning electron microscope (SEM). The types of functional groups, temperature frequency characteristics, low temperature performance and adhesion of SPMA were tested by the Fourier-transform infrared (FTIR) spectrometer, dynamic shear rheometer (DSR), bending beam rheometer (BBR) and contact angle meter. The results show that the SP and SCESP are rough and porous, and their main component is CaCO_3_, which is physically miscible to asphalt. When the loading frequency ranges from 0.1 Hz to 10 Hz, the complex shear modulus (G*) and phase angle (δ) of SPMA and SCESPMA increase and decrease, respectively. At the same load frequency, SCESPMA has a larger G* and a smaller δ than SPMA. At the same temperature, SCESPMA has a larger rutting factor (G*/sin δ) and better high-temperature deformation resistance than SPMA. SP and SCESP reduce the low-temperature cracking resistance of asphalt, of which SCESP has a more adverse effect on the low-temperature performance of asphalt than SP. When SP and SCESP are mixed with asphalt, the cohesion work (*Waa*), adhesion work (*Was*) and comprehensive evaluation parameters of water stability (*ER1*, *ER2* and *ER3*) of asphalt are improved. It is shown that both SP and SCESP have good water damage resistance, of which SCESP has better water damage resistance than SP. These research results have important reference value for the application of waste biological materials in asphalt pavement.

## 1. Introduction

The repeated load of vehicles can cause damage to the asphalt pavement. In order to improve the road performance of asphalt pavement, road workers have developed a variety of modifiers [1,2] to enhance the service quality and service life of asphalt pavement [3,4]. With the continuous improvement of environmental protection requirements, the concept of sustainable development has been introduced into road engineering, and more and more bio-based modified materials have been applied to asphalt pavement.

Bio-based modified materials have a series of advantages, such as regeneration, environmental protection and low price. Different bio-based modified materials have various effects on the road performance of asphalt [5]. At present, bio-based modified materials are mainly divided into bio-oil- and animal–plant-modified materials. They can modify some pavement performance of asphalt to obtain modified asphalt that is more beneficial to engineering application value.

The direct use of animals and plants after simple processing as modified asphalt materials has been gradually developed in recent years, and many scholars have conducted a series of researches in this field. Coconut seashell, coconut seashell fiber and palm seashell powders can increase the softening point temperature and rutting factor of asphalt [6,7,8]. Adding lignin and oil palm seashell powder to asphalt can significantly enhance the high-temperature stability of asphalt [9,10]. The addition of peanut seashell powder to asphalt can strengthen the rutting resistance of asphalt, and weaken its low-temperature performance and fatigue performance. When the content is 10%, the peanut seashell powder has better high-temperature storage stability [11]. Sargassum can promote high-temperature performance of asphalt. It has the best high-temperature stability when the content is 2.5%, and has good compatibility with asphalt [12]. Adding waste leather to 70 # asphalt improves its high-temperature performance, creep recovery ability and anti-aging performance [13]. Waste crayfish seashell powder can improve the high-temperature performance and creep resistance of asphalt [14]. Adding fish scales into neat asphalt can improve the adhesion, viscoelasticity, temperature sensitivity and permanent deformation resistance of asphalt [15]. The fish scale powder treated with silane coupling agent has little effect on the low-temperature and fatigue properties of asphalt, but it can change the viscoelastic components of asphalt, thereby improving the high-temperature stability of asphalt [16]. The addition of eggshell to asphalt mixture can reduce the amount of mineral powder and asphalt. At the same time, the road performance and mechanical property tests of modified asphalt are carried out. It is found that the high-temperature stability of asphalt is improved by the addition of eggshell, which can be applied to asphalt pavement engineering [17]. Some scholars have found that the eggshell powder has a rough, porous and loose structure, which improves the strange hardness and thermal stability. Its interaction with asphalt is a physical miscibility process [18]. When eggshell powder is mixed into asphalt, it makes asphalt hard and increases moisture susceptibility [19]. Laurel scallop, goby scallop and pectin scallop powders can significantly improve the elasticity of asphalt, reduce its low-temperature crack resistance, and keep the types of functional groups inside it unchanged [20]. The medium and low content of oyster powder can improve the low-temperature, rutting and fatigue properties of asphalt. This material can be used as a substitute for filler in asphalt mixtures [21]. The snail seashell powder can be well mixed with asphalt, which can reduce the void of asphalt mixture, thus improving the stability of asphalt mixture, but will reduce the permeability and rotary viscosity of asphalt [22]. Combined with the existing research, it is found that animal-modified materials can improve the high-temperature stability of asphalt to varying degrees. Some animal-modified materials can improve the fatigue performance of asphalt, but have negative effects on the low-temperature performance of asphalt to varying degrees. At the same time, most of these modified materials are physical modifications of asphalt.

China is rich in freshwater shellfish and aquatic products. Some seashell materials are currently used in handicraft manufacturing, decoration materials, agriculture and animal husbandry. These fields have high technical parameter requirements for seashell materials, and not all seashell products can meet these requirements. Therefore, a large number of seashells are directly dumped as garbage every year, resulting in great waste and environmental pollution. However, a large number of stone, asphalt and other materials are required in the construction process of asphalt pavement. If waste seashells can be used in asphalt pavement, the waste seashells pollution will be effectively reduced. At present, relevant scholars have been looking for countermeasures to reduce the environmental pollution caused by waste seashells. Existing researches have studied some waster seashells as modified asphalt materials. There is a lack of research on the road performance of Hyriopsis cumingii seashell powder-modified asphalt. Therefore, it is very meaningful to apply the Hyriopsis cumingii seashell powder to modify the asphalt, thus improving the road performance of asphalt pavement.

## 2. Materials Selection and Sample Preparation

### 2.1. Materials

#### 2.1.1. Aggregate and Asphalt

The limestone was used for the test and its surface energy was shown in Table 1. The A-70 # asphalt produced by SK Company in Korea was selected. According to the requirements of relevant specifications, the technical indexes of asphalt were tested [23,24], and its relevant properties are shown in Table 2.

#### 2.1.2. Seashell and Seashell Powder

The seashell of Hyriopsis cumingii was used in the test. Previous studies have shown that the seashell of Hyriopsis cumingii is divided into three layers. From the outside to the inside, it can be divided into the stratum corneum, prismatic layer and nacre, of which the prismatic layer with many holes is a porous structure, as shown in Figure 1 [25].

There are two kinds of seashell powders used in the test, that is, seashell powder (SP) and stratum corneum-exfoliated seashell powder (SCESP). The production process of the two is as follows: put the seashell and seashell with exfoliated stratum corneum into a small-scale high-speed multifunctional pulverizer for grinding, as shown in Figure 2, and then sift the powders through a 0.15 mm sieve to obtain the required ones, as shown in Figure 3 and Figure 4. The physical characteristics of shells and shell powder are shown in Table 3.

#### 2.1.3. Reagent for Contact Angle Test

The distilled water, glycerol and formamide were used in the test, and their surface energy parameters are shown in Table 4.

### 2.2. Sample Preparation

There have been studies on the use of biological materials to prepare modified asphalt. Generally, 5~30% of biological materials are used to prepare modified asphalt to study the road performance of modified asphalt [20,26]. The preparation of the seashell powder-modified asphalt sample in this study refers to the relevant existing research methods [20]. The sample preparation process is as follows: pour 2100 g neat asphalt into 7 sample cups evenly; heat the asphalt in the sample cup to 150 °C until it becomes complete flow; mix SP and SCESP into 6 sample cups, and stir the mixture for 30 min at 2500–3000 r/min under the high-speed shear instrument, as shown in Table 5. The seven modified asphalt samples were prepared as follows: neat asphalt (NA-70), 10% of SPMA (SPMA-10%), 20% of SPMA (SPMA-20%), 30% of SPMA (SPMA-30%), 10% of SCESPMA (SCESPMA-10%), 20% of SCESPMA (SCESPMA-20%), and 30% of SCESPMA (SCESPMA-30%).

### 2.3. Test Method

#### 2.3.1. Micro Characteristic Test

In order to study the appearance and chemical composition of SP and SCESP, in this study, the tungsten filament scanning electron microscope (SEM) Evo10 of Zeiss company in Oberkhein, Germany was used to observe the appearance and morphology of SP and SCESP, and the Bruker X-ray energy dispersive spectrometer (EDS) equipped on the instrument was adopted to analyze their main elements and obtain their main chemical components.

#### 2.3.2. Fourier-Transform Infrared (FTIR) Spectroscopy

The FTIR spectroscopy has important advantages in the analysis of organic compounds. It can be used for the analysis of organic compounds with asphalt and modified asphalt. In this study, the Bruker Tensor-27 FTIR spectrometer with scanning range of 500 cm^−1^–4000 cm^−1^ and scanning times of 64 was used to analyze the functional groups of seven asphalt samples. The FTIR spectrometer was produced by Bruker Corporation in Billerica, USA. Four groups of parallel tests were conducted on each asphalt sample. 

#### 2.3.3. Dynamic Shear Rheometer (DSR) Test

For purpose of study the effects of SP and SCESP on the high-temperature rheological properties of asphalt, the viscoelastic properties of SPMA and SCESPMA under different load frequencies were also investigated. The dynamic shear rheometer MCR302 of Anton Paar company in Graz, Austria was used to carry out temperature-frequency scanning test [27]. The loading frequency of frequency scanning test ranges from 0.1 Hz to 10 Hz, and the test temperatures include 58 °C, 64 °C, 70 °C and 76 °C. The complex shear modulus (G*) and phase angle (δ) of NA-70 and six modified asphalts were tested. The size of the sample is 25 mm in diameter and 1 mm in thickness. Four groups of parallel tests were carried out for each asphalt sample.

#### 2.3.4. Bending Beam Rheometer (BBR) Test

Referring to the relevant requirements of AASHTO 2015, the creep stiffness (*S*) and creep rate (*m*) of NA-70 and six modified asphalts were tested with the TE-BBR tester of Cannon Company in State College, USA, and it was required that the beam specimen at 60 s have a S ≤ 300 MPa and a m ≥ 0.3. The test temperatures of −12 °C, −18 °C and −24 °C were selected, and four groups of parallel tests were performed.

#### 2.3.5. Contact Angle Test

There is a close relationship between the water stability of asphalt and its adhesion and cohesion properties, and the adhesion and cohesion properties of asphalt can be obtained by contact angle test. As shown in Figure 5, the DSA100 contact angle meter of KRUSS Company in Hamburg, Germany was first used to measure the contact angle of asphalt sample and limestone. The distilled water, glycerol and formamide with known surface energy parameters were chosen as reagents. The interface diagram of reagents and limestone was shown in Figure 6. Five parallel tests were then carried out on each reagent and sample. If the data difference is large, the number of experiments will be supplemented. Finally, the average value of the five test results within the allowable error range was taken as the value of contact angle.

In order to evaluate the deformation resistance of asphalt and the adhesion between the asphalt and limestone, the cohesion and adhesion work indexes were introduced in this study. The calculation process is as follows.

The surface energy is the work required to produce a new interface per unit area of material at certain temperatures, and consists of a dispersion component and a polarity component, which can be expressed as Equation (1):(1)$$\gamma =\gamma^{d}+\gamma^{p}$$
where: γ is the surface energy; $\gamma^{d}$ is the dispersive component of the surface energy; and $\gamma^{p}$ is the polar component.

The cohesion work $W_{aa}$ refers to the deformation resistance of asphalt and modified asphalt slurry when relative displacement is generated under external force [28]. The greater the cohesion work, the stronger the deformation resistance. It can be expressed as Equation (2):(2)$$W_{aa}=2\gamma_{a}$$
where: $W_{aa}$ is the surface energy of asphalt or modified asphalt.

The adhesion characteristics between the asphalt (or modified asphalt) and limestone can be characterized by the adhesion work $W_{as}$. The greater the adhesion work, the better the stability of the two after combination. Its expression is as follows:(3)$$W_{as}=\gamma_{a}+\gamma_{s}-\gamma_{as}$$
where: $W_{as}$ is the adhesion work between the asphalt (or modified asphalt) and limestone; $\gamma_{a}$ is the surface energy of asphalt or modified asphalt; $\gamma_{s}$ is the surface energy of limestone; and $\gamma_{as}$ is the interface energy of asphalt (or modified asphalt) and limestone [29], which can be expressed as Equation (4).
(4)$$\gamma_{as}=\gamma_{a}+\gamma_{s}-2\sqrt{\gamma_{a}^{d}\gamma_{s}^{d}}-2\sqrt{\gamma_{a}^{p}\gamma_{s}^{p}}$$
where: $\gamma_{a}^{d}$ and $\gamma_{s}^{d}$ are dispersion components of asphalt (or modified asphalt) and limestone surface, respectively; $\gamma_{a}^{p}$ and $\gamma_{s}^{p}$ are polar components of asphalt (or modified asphalt) and limestone surfaces, respectively.

The Young Equation is shown in Equation (5).
(5)$$\gamma_{s}=\gamma_{as}+\gamma_{a}\cos\theta$$

The Young–Dupre Equation (6) can be obtained from Equations (1)–(5).
(6)$$W_{as}=\gamma_{a}\left(1+\cos\theta \right)=2\sqrt{\gamma_{a}^{d}\gamma_{s}^{d}}+2\sqrt{\gamma_{a}^{p}\gamma_{s}^{p}}$$

The water damage resistance of asphalt mixture can be characterized by the spalling work (*W_asw_*). The *W_asw_* is characterized by the external work required for the stripping of asphalt (or modified asphalt) and limestone under the action of water [30], which can be expressed by Equation (7).
(7)$$W_{asw}=W_{aw}+W_{sw}-W_{as}$$
where *W_aw_* is the adhesion work between the asphalt (or modified asphalt) and distilled water, and *W_sw_* is the adhesion work between the limestone and distilled water.

However, this index is not comprehensive. At present, scholars at home and abroad have found through research that the water damage resistance of asphalt and limestone is greatly related to the adhesion work, cohesion work and peeling work. Therefore, three water stability parameters *ER1*, *ER2* and *ER3* of asphalt and limestone were proposed [31]. The larger their values, the stronger the water damage resistance of asphalt and limestone, as shown in Equations (8)–(10).
(8)$$ER1=\left|\frac{W_{as}}{W_{asw}}\right|$$
(9)$$ER2=\left|\frac{W_{as}+W_{aa}}{W_{asw}}\right|$$
(10)$$ER3=\left|\frac{\min(W_{as},W_{aa})}{W_{asw}}\right|$$

## 3. Results and Discussion

### 3.1. Microscopic Characteristics of Seashell Powder

The SEM has unique advantages in observing the micro morphology of materials. Therefore, this study used this equipment to observe the appearance and chemical composition of SP and SCESP. The effects of appearance morphology and chemical composition of the two particles on the performance of NA-70 were explored, and the test results are shown in Figure 7 and Table 6.

From Figure 7, it can be seen that the surface of SP and SCESP are rough with more folds, unevenness, internal pores and porosity. The SCESP has more internal holes and pores than the SP. It has been found that there are many holes in the prismatic layer of the seashell, which may be closely related to the existence of a large number of holes in SP and SCESP. There is more contact area between the SP (or SCESP) and NA-70 after high-speed shear stirring, which helps form a strong adsorption capacity between them.

Table 5 shows that the main elements of SP and SCESP are C, O, CA, Si, S, Al and Mg, and the main compounds are CaCO_3_, SiO_2_, Al_2_O_3_ and MgO; the content of CaCO_3_ is the highest and the composition is similar to that of limestone, which indicates that when SP and SCESP are added to NA-70, CaCO_3_ plays a major role in the change in road performance of NA-70.

### 3.2. Functional Group Analysis

There are many functional groups in asphalt [32]. In order to analyze the effects of SP and SCESP on the functional groups of NA-70, this study used the FTIR spectroscopy to analyze the functional groups of SPMA and SCESPMA, and the test results are shown in Figure 8.

It can be seen from Figure 8 that in the range of 4000–500 cm^−1^, NA-70, SPMA and SCESPMA have the same absorption peak. There are characteristic peaks caused by stretching vibration near 2924 cm^−1^ and 2853 cm^−1^, C=C stretching vibration characteristic peaks near 1600 cm^−1^, aliphatic C-H bending deformation near 1456 cm^−1^ and 1376 cm^−1^, sulfoxide group (S=O) absorption peaks near 1030 cm^−1^, and C-H bond bending vibration near 860 cm^−1^, 813 cm^−1^, 746 cm^−1^ and 722 cm^−1^. It is shown that no new chemical components are generated after the two are mixed, which is physically miscible. Therefore, the modification of SP and SCESP to NA-70 is physical.

### 3.3. Analysis of High-Temperature Rheological Properties

In order to study the effect of frequency on their viscoelasticity, the frequency sweep of SPMA and SCESPMA was carried out to obtain G* and δ. The curves with frequency are shown in Figure 9 and Figure 10, respectively.

It can be seen from Figure 9 that the G* of SPMA and SCESPMA increases linearly with the increase in load frequency at 58 °C~88 °C. The specific rules are: SCESPMA-30% > SPMA-30% > SCESPMA-20% > SPMA-20% > SCESPMA-10% >SPMA 10% > NA-70. When the load frequency ranges from 0.1 Hz to 10 Hz, the G* of SPMA and SCESPMA increases; when the temperature increases, the G* of SPMA and SCESPMA decreases. It is shown that the load frequency has an important influence on the viscoelasticity of SPMA and SCESPMA. The addition of SPMA and SCESPMA can improve the load resistance of asphalt. The SCESP has better deformation resistance, indicating that the stratum corneum of seashell has an adverse effect on the deformation resistance of asphalt. In addition, SPMA is more sensitive to temperature.

It can be seen from Figure 10 that at 58 °C~76 °C, the δ of NA-70, SPMA and SCESPMA gradually decreases with the increase in frequency. At 0.1 Hz~1.59 Hz, the value of δ decreases significantly, while at 1.59 Hz~10 Hz, the value of δ decreases slowly. The specific rules are: NA-70 > SCESPMA-10% > SCESPMA-20% > SPMA-10% > SCESPMA-30% > SPMA-20% > SPMA-30%. It is indicated that the temperature has a significant effect on the viscoelasticity of SPMA and SCESPMA; as the temperature increases, the effect is more significant. At low frequency, SPMA and SCESPMA show more viscous performance, while with the increase in frequency, the viscous performance decreases, and the elastic performance increases.

To analyze the high-temperature deformation resistance of SPMA and SCESPMA, the rutting factors (G* and sin δ) at 58 °C, 64 °C, 70 °C and 76 °C were measured at 1.59 Hz, and the results are shown in Figure 11.

From Figure 11, it can be seen that the G* and sin δ of NA-70, SPMA and SCESPMA gradually decrease from 58 °C to 76 °C, and their specific rules are: SCESPMA-30% > SPMA-30% > SCESPMA-20% > SPMA-20% > SCESPMA-10% > SPMA-10% > NA-70. It is shown that the temperature has significant effect on the G* and sin δ of SPMA and SCESPMA, which becomes less and less significant with the increase in temperature. SCESPMA has better high-temperature rutting resistance than SPMA. Previous studies have analyzed the impact of bio-based modified materials on the high-temperature performance of asphalt, and few have analyzed the impact of different components of these materials on the G*/sin δ of asphalt [11]. The effects of SP and SCESP on the G*/sin δ of asphalt were analyzed, respectively, in this study, and the potential of seashell materials to improve the high-temperature performance of asphalt was studied in detail.

The U.S. Strategic Highway Research Program (SHRP) uses the rutting factors G* and sin δ as rating indexes for the high-temperature performance of asphalt [33,34]. In this study, a continuous high-temperature performance gradation (PG) analysis was conducted for SPMA and SCESPMA with reference to ASTM D 7643-16, and the specific results are shown in Figure 12.

As shown in Figure 12, the temperature T_cH_ of continuous high-temperature PG of NA-70 increases as the contents of SP and SCESP increase. Compared with that of NA-70, the T_cH_ of SCESPMA-10%, SCESPMA-20% and SCESPMA-30% increases by 4.49%, 6.98% and 9.49%, respectively, and the T_CH_ of SPMA-10%, SPMA-20% and SPMA-30% increases by 4.22%, 6.21% and 8.52%, respectively.

It can be found that the temperature T_CH_ of SCESPMA is higher than that of SPMA at the same content. It is indicated that SCESP improves the high-temperature performance of NA-70, which may be related to the prismatic layer of seashell with more holes. The holes increases the contact surface between the seashell powder and NA-70, thus improving its high-temperature rutting resistance.

### 3.4. Analysis of Rheological Properties at Low Temperature

The evaluation parameters of low-temperature performance of asphalt are usually *S* and *m* [35,36]. In this study, the *S* and *m* values of NA-70 and SPMA and SCESPMA were tested, and the values of *S* and *m* are shown in Figure 13. To analyze the low-temperature cracking resistance of NA-70, SPMA and SCESPMA more comprehensively, this study used the comprehensive evaluation parameter of low-temperature performance λ=S/m (the smaller the λ value, the better the low temperature performance), and the values of λ are shown in Figure 14.

It can be observed from Figure 13a that the *S* value of SCESPMA are larger than that of SPMA at the same temperature; with the increase in SP and SCESPMA content, the *S* value of NA-70 increases. The results show that the low-temperature bending creep performance of NA-70 with SP and SCESPMA added is worse, and the low-temperature bending creep performance of SPMA is better than that of SCESPMA at the same content.

It is obvious from Figure 14 that the λ value of SPMA is smaller than that of SCESPMA at the same temperature, and the λ values of both SPMA and SCESPMA increase as the contents of SP and SCESP increase. It is indicated that SPMA has better low-temperature crack resistance than SCESPMA. Previous studies have found that adding other seashell materials to asphalt will generally reduce its low-temperature crack resistance [20,21], which is consistent with the results of this study. However, this study finds that the stratum corneum of seashell has better low-temperature crack resistance than other structural layers.

In this study, NA-70, SPMA and SCESPMA were analyzed by using the index T_cL_ of continuous low-temperature PG with reference to ASTM D 7643-16, and the values of T_cL_ are shown in Figure 15.

In Figure 15, it is shown that the temperature T_cL_ of continuous low-temperature PG of NA-70 increases with the increase in SP and SCESP contents; compared with that NA-70, the T_cL_ of SPMA increases by 5.52%, 16.72%, and 28.44%, respectively, and the T_cL_ of SCESPMA increases by 6.51%, 18.73% and 30.10%, respectively.

### 3.5. Analysis of Water Stability

To verify the reliability of contact angle test results, after obtaining the contact angles of NA-70, SPMA and SCESPMA with three reagents, a linear fitting was performed on the surface energy $\gamma_{l}$ and $\gamma_{l}\cos\theta$ of different reagents, and the test results are shown in Figure 16.

According to Figure 16, $\gamma_{l}$ and $\gamma_{l}\cos\theta$ of NA-70 and SPMA and SCESPMA show good linear correlation above 0.89, indicating that the contact angle test data are reliable.

Based on the data in Table 1 and Table 3, *W_aa_*, *W_as_* and *W_asw_* of NA-70, SPMA and SCESPMA were obtained by Equations (2), (3) and (7), and the test results are shown in Figure 17.

As shown in Figure 17, the specific rules of *Waa* and *Was* of SPMA and SCESPMA are: SCESPMA-30% > SPMA-30% > SCESPMA-20% > SPMA-20%> SCESPMA-10% > SPMA-10% > NA-70. It is found that compared with NA-70, SCESPMA improves *Waa* and *Was* by 9.91~27.70% and 5.44~15.04%, respectively, and SPMA enhances *Waa* and *Was* by 7.08~26.23% and 4.26~14.51%, respectively. The specific rules of *Wasw* of SPMA and SCESPMA are: NA-70 > SCESPMA-10% > SPMA-10% > SCESPMA-20%> SPMA-20%> SCESPMA-30% > SPMA-30%. It is shown that compared with NA-70, the *Wasw* of SCESPMA and SPMA is reduced by 0.38~1.83% and 0.72~2.1%, respectively.

The results show that both SP and SCESP improve the *Waa* and *Was* of NA-70 except for the *W_asw_*. SCESPMA has better adhesion and cohesion than SPMA, indicating that the stratum corneum in the seashell is not conducive to improving the adhesion and cohesion of NA-70. The existing research on bio-based modified asphalt materials mainly focuses on its high- and low-temperature rheological properties, micro morphology and chemical composition [15,37], and rarely involves the adhesion and cohesion properties of seashell powder-modified asphalt. The results of this study show that the water stability of asphalt and limestone is not only related to *W_asw_*, but also has a close relationship with *W_aa_* and *W_as_*, and the above three indexes should be considered comprehensively in this study. Hence, this study introduced the comprehensive evaluation parameters *ER1*, *ER2* and *ER3* for asphalt and limestone, so as to perform a comprehensive evaluation on the water stability of SPMA, SCESPMA and limestone, and the test results are shown in Figure 18.

It can be seen from Figure 18 that the specific rules of *ER1*, *ER2* and *ER3* of SPMA and SCESPMA are: SCESPMA-30% > SPMA-30% > SCESPMA-20% > SPMA-20%> SCESPMA-10% > SPMA-10% > NA-70; as the contents of SP and SCESP increase, *ER1*, *ER2* and *ER3* increase; and at the same content, the *ER1*, *ER2* and *ER3* values of SCESPMA are larger than those of SPMA. It shows that SP and SCESP can improve the water damage resistance of NA-70 and limestone, and SCESP is more conducive to improving the water damage resistance of NA-70 than SP, which indicates that the stratum corneum in the seashell is not conducive to the water stability of asphalt and limestone.

## 4. Conclusions

In this study, the potential of SP and SCESP as asphalt modifiers was discovered, and the high-low temperature performance and water stability of SPMA and SCESPMA were studied. The main conclusions can be summarized as follows.

(1)The surface of SP and SCESP is rough with more pores and holes, and their main component is CaCO_3_. SCESP has more pores and holes than SP, and has better adsorption effect during mixing with asphalt, thus giving the modified asphalt a better adhesion effect. From the perspective of the type and content of characteristic functional groups of modified asphalt. The modification of SP and SCESP to NA-70 is a physical process.(2)Temperature–frequency tests show that SP and SCESP improve the high-temperature deformation resistance of NA-70. The G* of SPMA and SCESPMA is linear with the load frequency. At the same temperature, the G* of SCESPMA is more sensitive to the load frequency than that of SPMA, and SCESP has better high-temperature improvement effect on NA-70 than SP.(3)BBR tests indicate that SP and SCESP increase the creep modulus (S) and reduce the creep rate (m) of NA-70, resulting in a decrease in the low-temperature crack resistance of NA-70, and SCESP is more detrimental to the low temperature crack resistance of NA-70 than SP at the same content.(4)The contact angle test shows that SP and SCESP enhance the cohesion work, adhesion work and water stability of NA-70; in addition, at the same content, SCESP strengthens the cohesion work, adhesion work and water stability of NA-70 better than SP.

## Figures and Tables

**Figure 1 materials-15-06788-f001:**
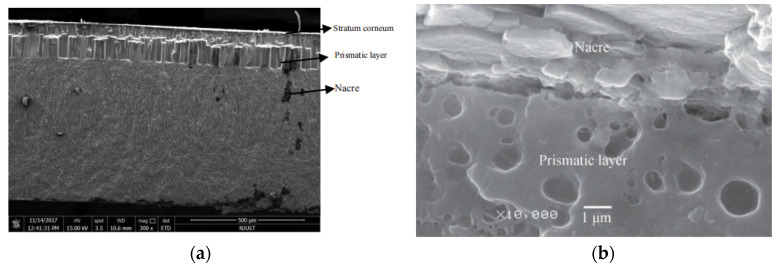
Microstructure of Hyriopsis cumingii seashell. (**a**) Distribution of seashell structure layer; (**b**) Distribution of holes in the seashell.

**Figure 2 materials-15-06788-f002:**
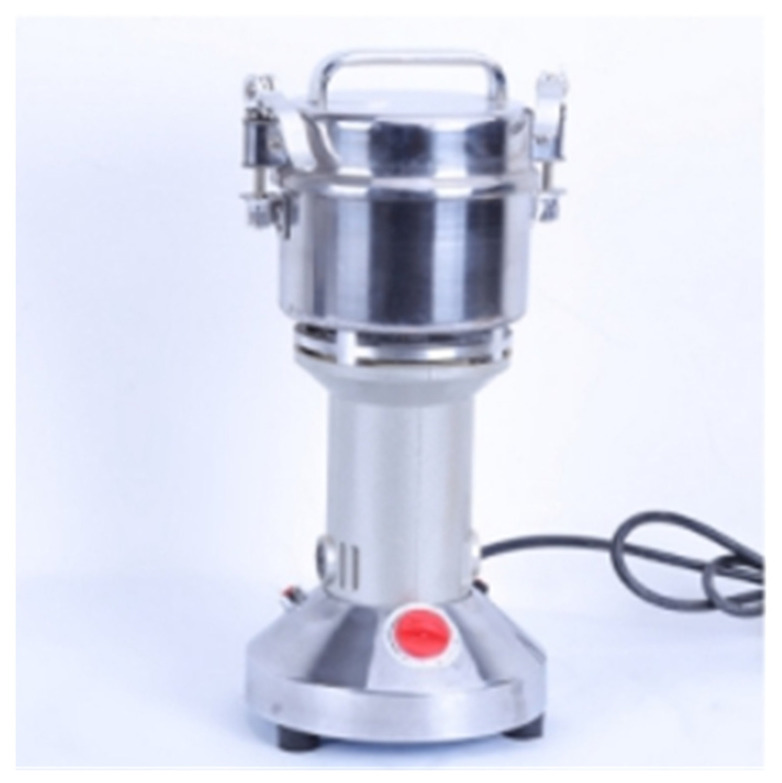
Small high-speed multifunctional pulverizer.

**Figure 3 materials-15-06788-f003:**
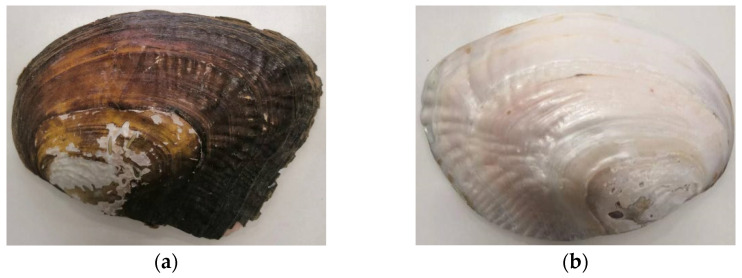
Seashell and stratum corneum-exfoliated seashell. (**a**) Seashell; (**b**) Stratum corneum-exfoliated seashell.

**Figure 4 materials-15-06788-f004:**
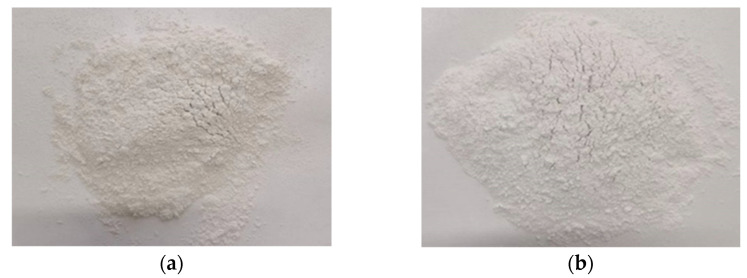
Seashell powder and stratum corneum-exfoliated seashell powder; (**a**) seashell powder; (**b**) Stratum corneum-exfoliated seashell powder.

**Figure 5 materials-15-06788-f005:**
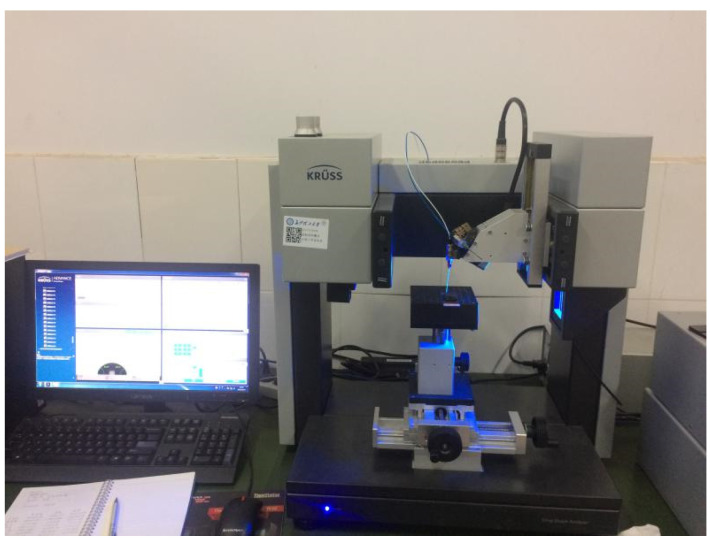
Contact angle meter.

**Figure 6 materials-15-06788-f006:**
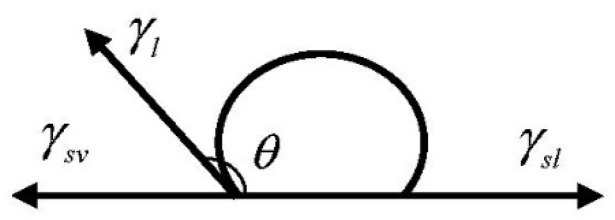
Schematic diagram of interface between the reagent and limestone.

**Figure 7 materials-15-06788-f007:**
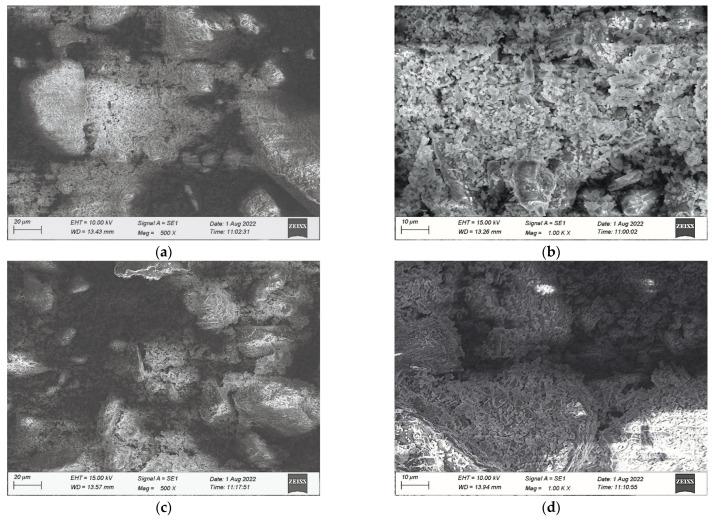
Microscopic images of SP and SCESP. (**a**) SCESP × 500; (**b**) SCESP × 1000; (**c**) SP × 500; (**d**) SP × 1000.

**Figure 8 materials-15-06788-f008:**
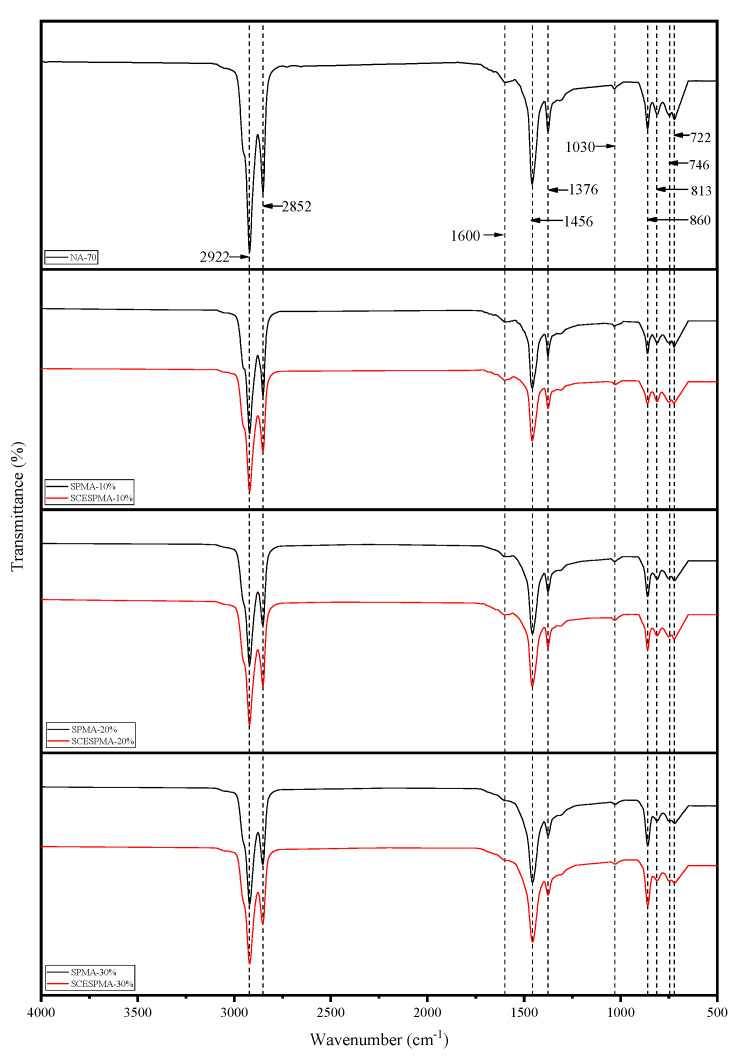
Functional groups of SPMA and SCESPMA.

**Figure 9 materials-15-06788-f009:**
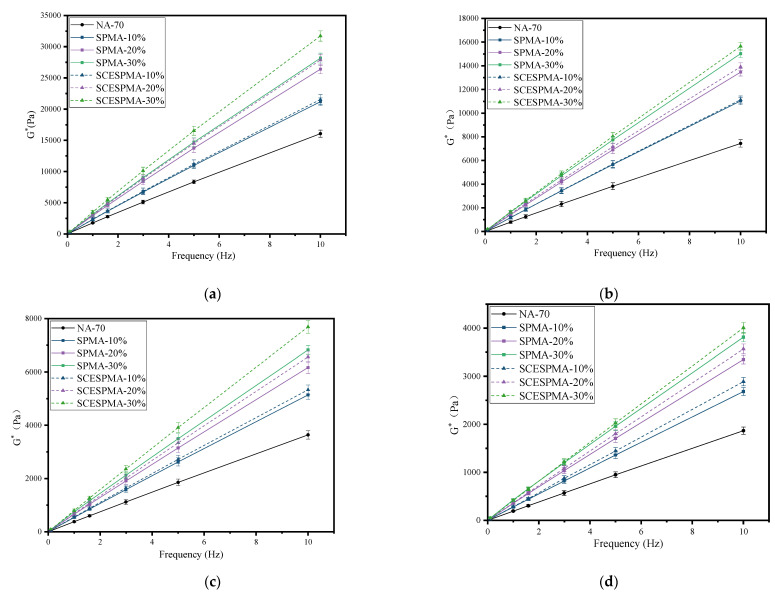
Relationship among G * and frequency of SPMA and SCESPMA at different temperatures. (**a**) 58 °C; (**b**) 64 °C; (**c**) 70 °C; (**d**) 76 °C.

**Figure 10 materials-15-06788-f010:**
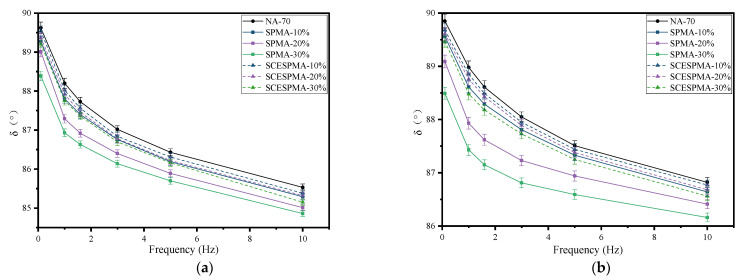
Relationship among δ and frequency of SPMA and SCESPMA at different temperatures. (**a**) 58 °C; (**b**) 64 °C; (**c**) 70 °C; (**d**) 76 °C.

**Figure 11 materials-15-06788-f011:**
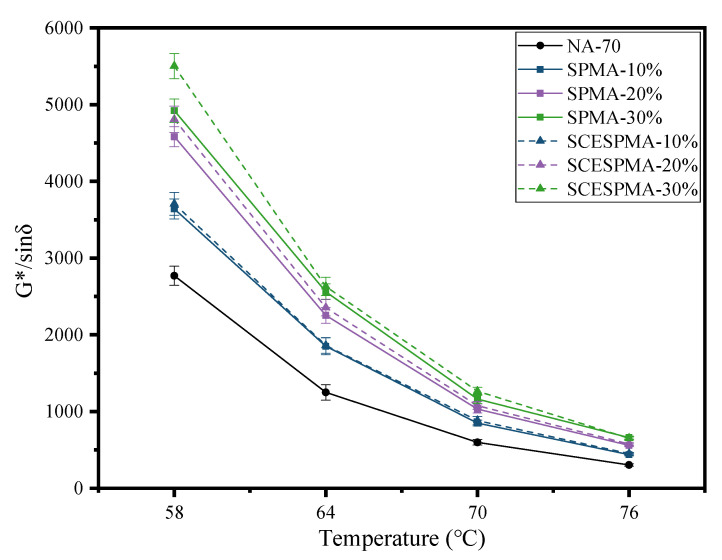
G* and sin δ of SPMA and SCESPMA at different temperatures.

**Figure 12 materials-15-06788-f012:**
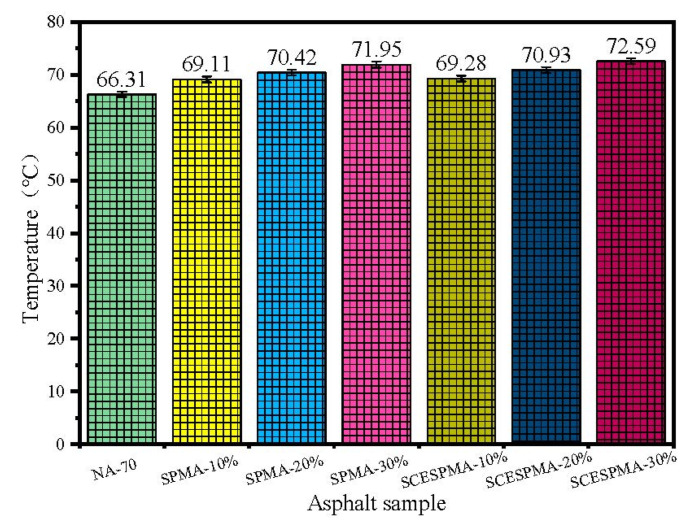
Continuous high-temperature PG analysis of SPMA and SCESPMA.

**Figure 13 materials-15-06788-f013:**
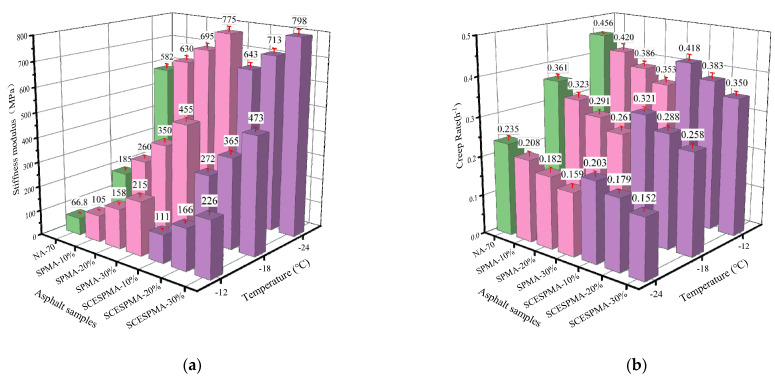
*S* and *m* of SPMA and SCESPMA at different temperatures. (**a**) *S*; (**b**) *m*.

**Figure 14 materials-15-06788-f014:**
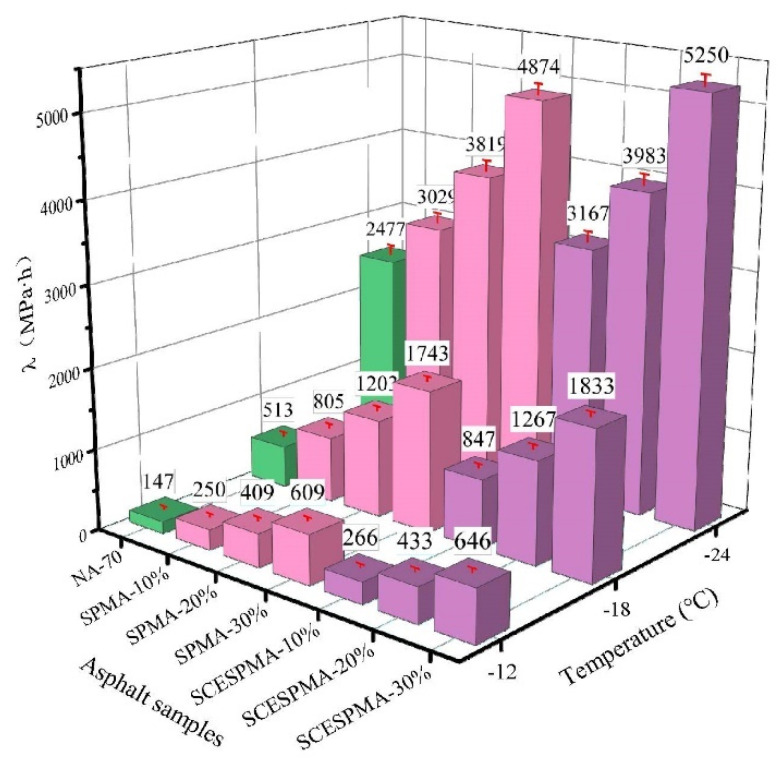
λ of SPMA and SCESPMA at different temperatures.

**Figure 15 materials-15-06788-f015:**
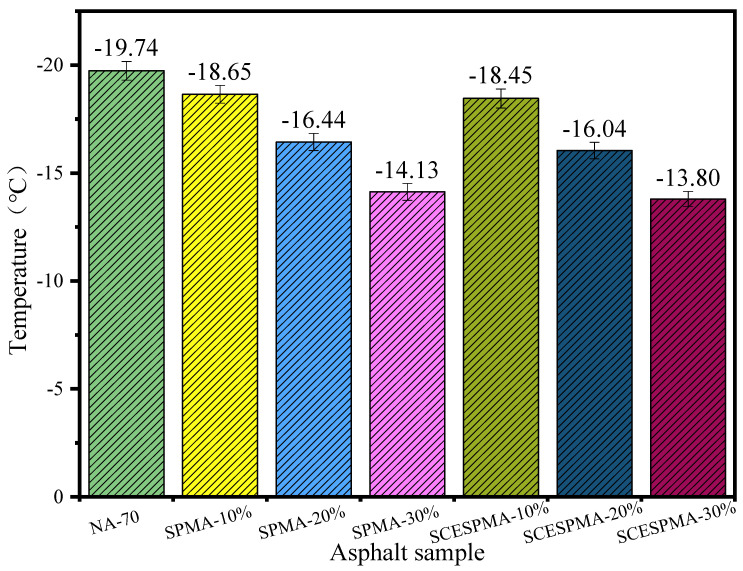
Continuous low-temperature PG analysis of SPMA and SCESPMA.

**Figure 16 materials-15-06788-f016:**
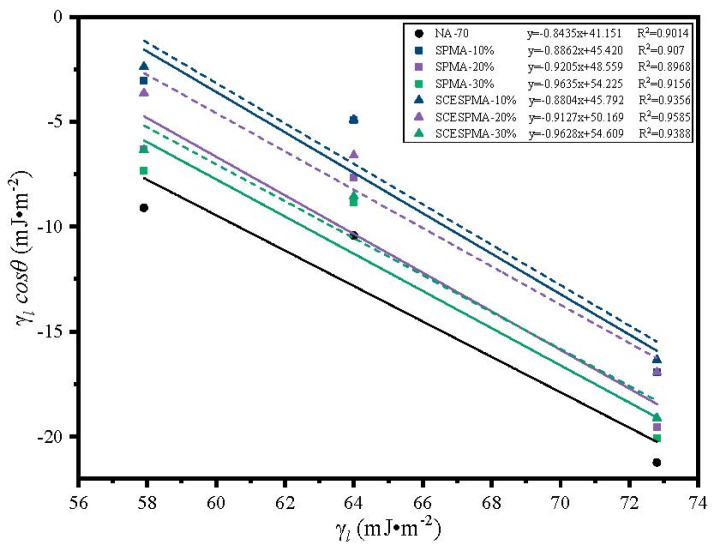
Relationship between *γ_l_* and *γ_l_cosθ*.

**Figure 17 materials-15-06788-f017:**
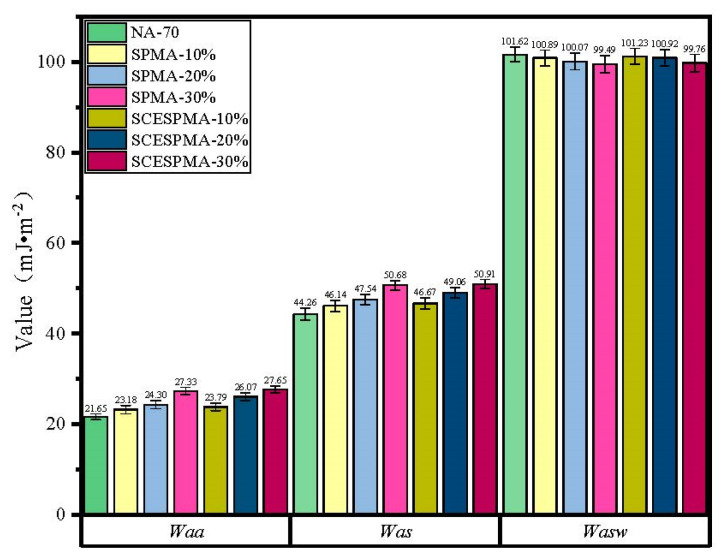
Adhesion properties of SPMA and SCESPMA.

**Figure 18 materials-15-06788-f018:**
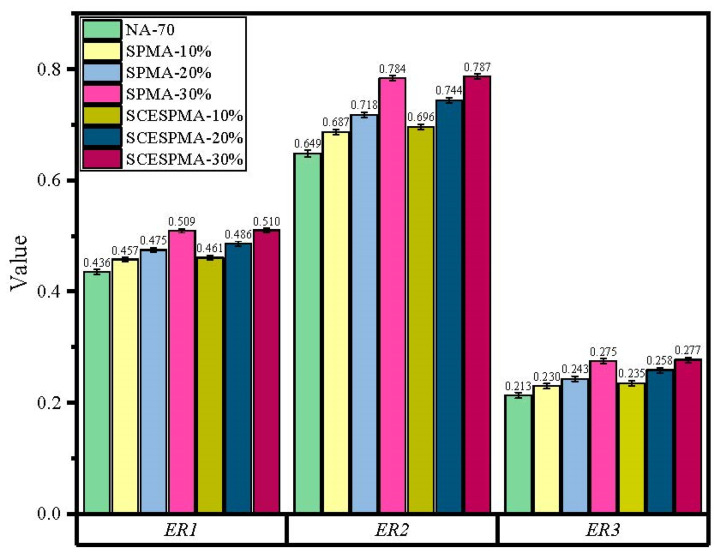
Comprehensive water stability evaluation parameters of SPMA and SCESPMA.

**Table 1 materials-15-06788-t001:** Surface energy parameters of limestone.

Aggregate	$\gamma_{s}/\mathrm{mJ}\cdot m^{-2}$	$\gamma_{s}^{d}/\mathrm{mJ}\cdot m^{-2}$	$\gamma_{s}^{p}/\mathrm{mJ}\cdot m^{-2}$
Limestone	48.76	43.66	5.10

**Table 2 materials-15-06788-t002:** Test results of A-70 # neat asphalt.

Items		Test Specification: JTG E20-2011		
		Measured Values	Technical Requirements	Reference Methods
Penetration (25 °C, 100 g, 5 s) (0.1 mm)		69.9	60~80	T0604-2011.
Penetration index (PI)		0.201	−1.5~+1.0	T0604-2011.
Length (5 cm/min, 10°C) (cm)		33	≥15	T0605-2011.
Length (5 cm/min, 15°C) (cm)		>100	≥100	T0605-2011.
Softening point (Global method) (°C)		49.5	≥46	T0606-2011.
Wax content (distillation) (%)		0.5	<2.2	T0615-2011.
Flash point (°C)		335	≥260	T0611-2011.
Solubility (%)		99.87	≥99.5	T0607-2011.
Density (25 °C) (g/cm)^3^)		1.037	Measured	T0603-2011.
60 °C Dynamic viscosity (Pa·s)		199.8	≥180	T0620-2000.
Rolling thin film oven test (163 °C, 85 min)	Mass loss (%)	−0.065	≤±0.8	T0609-2011.
	Residual penetration ratio (%)	81	≥61	T0604-2011.
	Residual ductility (10 °C)	7	≥6	T0605-2011.
	Residual ductility (15 °C)	28	≥15	T0605-2011.

**Table 3 materials-15-06788-t003:** Physical characteristics of shell and shell powder.

Seashell Name	Seashell Structure	Types of Seashell Powder	Appearance	Particle Size of Seashell Powder
Hyriopsis cumingii	Stratum corneum Prismatic layer Nacre	Seashell powder	Light yellow powder	<0.15 mm
		Stratum corneum-exfoliated seashell powder	Milky white powder	

**Table 4 materials-15-06788-t004:** Surface energy parameters of reagents.

Reagents	$\gamma_{l}$/mJ·m^−2^	$\gamma_{l}^{d}$/mJ·m^−2^	$\gamma_{l}^{p}$/mJ·m^−2^
Distilled water	72.8	21.8	51
Glycerol	64	34	30
Formamide	58	39	19

**Table 5 materials-15-06788-t005:** Contents of SP and SCESP in modified asphalt.

Sample No.	1	2	3	4	5	6	7
Content of SP (g)	0	10	20	30			
Content of SCESP (g)	-	-	-	-	10	20	30

**Table 6 materials-15-06788-t006:** Main components of SP and SCESP.

Sample Type	Main Components						
SP	Element	C	O	Ca	Si	S	Al
	Weight (%)	42.54	37.80	12.39	3.86	1.85	1.56
	Chemical compound	CaCO_3_		SiO_2_		Al_2_O_3_	
	Weight (%)	76.32		15.25		8.43	
SCESP	Element	C	O	Ca	Si	S	Mg
	Weight (%)	33.25	44.55	18.58	1.85	0.91	0.86
	Chemical compound	CaCO_3_		SiO_2_		MgO	
	Weight (%)	86.62		8.23		5.15	

## Data Availability

Data available on request due to restrictions e.g., privacy or ethical.

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
