# Peer review of "Analysis of the Influence of Waste Seashell as Modified Materials on Asphalt Pavement Performance"

_materials, 2022, doi:10.3390/ma15196788_

Round 1
Reviewer 1 Report
file name "Review Report"

Author Response
感谢您的认真审阅和宝贵意见。这些意见对我们论文的修改和完善很有帮助,对我们的研究工作具有重要的指导意义。我们仔细研究了所有评论,并进行了更正。我们希望能得到您的认可。本稿件的主要更正及对审稿人意见的回复请见附件。

Reviewer 2 Report
The authors analyzed the properties of asphalt modified by shell powder through various tests. In general, the manuscript has good logic and organization. More background information is suggested in the experiment design and material testing section. More references are suggested on the motivation and experiment. In Section 2, more detailed information could be given to explain the reason for the design, choice of equipment, test methods, etc. A small paragraph for each subsection seems pale. In Section 3, the authors could validate the results with others' work in the literatures and analyze the advantages and challenges of the results. More information is suggested in the conclusions.
Author Response
感谢您的认真审阅和宝贵意见。这些意见对我们论文的修改和完善很有帮助,对我们的研究工作具有重要的指导意义。我们仔细研究了所有评论,并进行了更正。我们希望能得到您的认可。 本稿件的主要更正及对审稿人意见的回复请见附件。

Round 2
Reviewer 1 Report
Only comment 4, I think the authors do not understand my point.
I suggest the authors to rewrite the introduction based on previous studies related to shell powder instead of bio-oil.
I know that seashells consider as bio-materials. But the effects of the two boi-materials are different for example shell powder increases the rutting resistance. Meanwhile, the bio-oil lower the rutting resistance so I suggest the authors to add a more in-depth review of different kind of shells ((only)) (not bio-oil).
and I suggest the author to add a discussion on Fig 17.
My regards...
Author Response
We have made modifications according to your requirements. Please see the attachment for specific modifications.
